# Monitoring the health of wolves (*Canis lupus*): Integrating conservation and public health

**Elisabetta Ferraro**[1,2]*, **Graziana Da Rold**[1,2], **Roberto Celva**[2], **Elisa Dalla Libera**[2], **Stefania Leopardi**[2], **Giulia Simonato**[1], **Paola De Benedictis**[2], **Nadia Cappai**[3], **Arianna Dissegna**[4], **Carlo Vittorio Citterio**[2], **Rudi Cassini**[1], **Federica Obber**[2]

1 Department of Animal Medicine, Production and Health, University of Padova, Legnaro, Italy, 2 Istituto Zooprofilattico Sperimentale delle Venezie, Legnaro, Italy, 3 Foreste Casentinesi National Park, Pratovecchio-Stia, Italy, 4 Department of Biology, University of Padova, Padova, Italy

* elisabetta.ferraro.1@phd.unipd.it

## Abstract

The grey wolf (*Canis lupus*) population is expanding in parts of Europe due to legal protection and favorable ecological conditions. As wolves increasingly move into urban and suburban areas, interactions with domestic dogs become more frequent, raising the risk of pathogen transmission and posing potential threats to both wolf conservation and public health. This study investigated the health status of wolves in the Foreste Casentinesi National Park (Italy) using non-invasive fecal sampling conducted between May 2019 and March 2020. Samples were genetically analyzed to identify individuals and then screened for viral pathogens, Canine Coronavirus and Parvovirus, using PCR, Sanger sequencing, and phylogenetic analysis. Parasitological examinations were performed using flotation techniques on whole samples, and real-time PCR targeting *Echinococcus granulosus* and *E. multilocularis* was conducted on selected samples. Of the 260 samples collected, genetic analysis identified 80 individual wolves belonging to 8 packs. Only one sample tested positive for Coronavirus (1.2%), and none for Parvovirus. The detected sequence clustered with strains previously reported in wolves and foxes in Italy. Copromicroscopy revealed a high prevalence of veterinary-relevant endoparasites, including *Eucoleus* spp. (90.0%), *Sarcocystis* spp. (42.5%), Taeniids (28.7%), and Ancylostomatids (26.2%). *Trichuris vulpis*, *Toxocara canis*, and coccidia showed prevalence rates below 2%. All 104 samples tested for *E. granulosus* or *E. multilocularis* were negative. These findings suggest that while wolves in the FCNP commonly harbor several canine parasites, their role in the transmission of zoonotic pathogens appears limited. Although phylogenetic data suggest that coronavirus strains tend to cluster within wildlife species, molecular data on domestic dogs remain scarce. Nonetheless, the high prevalence of shared parasites highlights the need for ongoing surveillance in both wild canids and domestic carnivores. As wolves increasingly inhabit human-dominated

**Data availability statement:** The sequence of CCoV was deposited in GenBank under accession number PV661850 (https://www.ncbi.nlm.nih.gov/genbank).

**Funding:** The author(s) received no specific funding for this work.

**Competing interests:** The authors have declared that no competing interests exist.

landscapes, understanding disease dynamics at the wildlife–domestic interface is essential for effective conservation and public health strategies.

## Introduction

The wolf (*Canis lupus* Linnaeus, 1758) is a large carnivore, native to Europe, and an integral part of the continent's biodiversity and natural heritage. Despite this, it was driven to near extinction across Europe, reaching its lowest population levels during the 1960s and 1970s. However, small and fragmented populations survived in the southern European peninsulas and parts of Eastern Europe [1]. Among these, the Apennine wolf subpopulation (*Canis lupus italicus*, Altobello 1921) is notable for having never disappeared completely, acting as an ecological link between the southern Italian population and the Western Alps, thereby facilitating gene flow among metapopulations [2]. Legal protection (Habitat Directive 92/43/EEC and Italian Law 157/1992), combined with socioecological changes such as rural-to-urban migration, reforestation, and an increase in ungulate populations, has enabled the re-expansion of wolves across Europe, laying the groundwork for a large-scale recovery of this species. As wolves expand into new environments, some individuals are recognized to disperse into highly anthropized habitats, both urban and suburban, which are considered major drivers of disease exposure for wolves in Europe and North America [3], as these areas increase the frequency of interaction among wolves and domestic animals, including dogs. Indeed, dog-wolf interactions, such as mating, predatory events or resource sharing, could facilitate the transmission of infectious and parasitic diseases. This dynamic may pose a threat to wolf conservation and public health, as canine pathogens may spill over into wolves [4–6], which may then act as reservoirs or carriers, spreading diseases back to both domestic animals and humans (e.g., canine distemper virus and *Echinococcus granulosus*) [3].

Transmissible diseases in wildlife can impact the mortality rates of wild animal populations, ranging from mild effects that contribute to natural density control to severe consequences, including, in extreme cases, population extinction [6,7]. However, accurately assessing the real risks can be challenging [7]. Among the pathogens that threaten the health of wolf populations, Carnivore Protoparvovirus 1 (CPPV-1) and Canine Enteric Coronavirus (CCoV) play a major role. Both are enteric pathogens, and their impact may vary depending on the age, susceptibility, and comorbidities of infected individuals [8–10]. CPPV-1 includes different genogroups, among which Canine Parvovirus 2 (CPV-2) and its variants (CPV-2a, CPV-2b, CPV-2c) are the etiological agents of hemorrhagic gastroenteritis, an extremely contagious disease with high morbidity and mortality that affects canids, felids, mustelids, and procyonids, and this wide host range contributes to its global spread [10,11]. CCoV is classified into two serotypes, CCoV-I and CCoV-II, based on spike proteins, each exhibiting different serological properties [12]. It is a cause of usually mild but highly contagious enteritis in puppies under 12 weeks of age, particularly in overcrowded environments such as breeding facilities and kennels [13]. In some cases, the disease can be fatal, especially in animals co-infected with other pathogens, such as CPV-2, canine

adenovirus type 1 or canine distemper virus [9,12]. Recently, pantropic CCoV-IIa serotypes (pCCoV) have emerged in dogs, capable of causing systemic infections characterized by fever, hemorrhagic gastroenteritis, neurological signs, and leukopenia, potentially leading to death even in the absence of co-infections [8,9,12,14]. The primary route of transmission for both viruses is direct contact with feces or contaminated materials, as they are highly resistant in the environment [10,15,16]. Dogs can play a significant role in the spread of CPV-2 in wild populations, as feral, stray or unvaccinated dogs may act as carriers. Once the viruses are introduced into the environment, transmission between dogs and wild canids can occur through contact with scats, predation, coprophagia, or social behaviors [11]. Additionally, the persistence of infection among wolves is supported by their social structure and behaviors, which facilitate fecal-oral transmission within the population [15].

Over the past decades, numerous studies have investigated parasitic infections in the Italian wolf population, with a particular emphasis on tapeworms and *Echinococcus* spp. due to their zoonotic relevance [17–21]. *Echinococcus granulosus*, the causative agent of cystic echinococcosis, is primarily associated with a domestic transmission cycle, in which sheep play a central role due to their high population density and high cyst fertility rates. The domestic dog serves as the main definitive host. Currently, there is no clear evidence supporting the existence of a true sylvatic cycle of *E. granulosus* sensu stricto in Southern Europe that is sustained independently of domestic dogs and livestock. In contrast, *Echinococcus multilocularis,* the agent of the alveolar echinococcosis, primarily circulates between rodents (mainly voles) and canids that prey on them. Red foxes (*Vulpes vulpes* L., 1758) are the principal definitive hosts, due to their high infection rates, feeding habits, and population densities. Domestic dogs, although highly susceptible to infection, show low prevalence due to limited rodent predation and occasional deworming. Wolves have been identified as hosts in some regions, but their dietary preference for large ungulates limits their role in the parasite's life cycle [22].

In Central Italy, research has primarily focused on the role of wolves in the transmission cycle of *Echinococcus granulosus* sensu stricto (G1-G3), suggesting a potential link to the domestic (sheep-dog) cycle rather than the presence of a strictly sylvatic one [21]. This hypothesis is supported by the low prevalence of fertile cysts in wild ungulates [19], the relatively low infection rates of *E. granulosus* detected in wolves [21], and the significantly higher prevalence observed in adult sheep in Central and Southern regions of Italy [23]. Other species of the *E. granulosus* sensu lato complex were rarely reported in Italy [24], while *E. multilocularis* is distributed in Central and North-eastern Europe with a single historical focus of circulation in Italy, in the Northern area of Bolzano province [25]. Notably, Massolo et al. (2018) [26] unexpectedly detected eggs of *E. multilocularis* and *E. ortleppi* (G5) in wolves from North-western Italy, underscoring the need for continued surveillance and investigation. Furthermore, wolves share many species of intestinal parasites with domestic dogs and other wild canids, with potential inter-specific transmission. In conclusion, ongoing monitoring remains crucial to further elucidate the epidemiological role of wolves in the transmission cycle of all these parasites.

The aim of this study was to broaden the investigation of the health status of the wolf population within the Foreste Casentinesi, Monte Falterona e Campigna National Park (FCNP). This was achieved by integrating population monitoring, focused on abundance estimates and pack identification (for further details, see Dissegna et al., 2023 [27]), with a comprehensive health survey using non-invasive sampling methods. Specifically, we conducted a survey on the presence of parasitic and viral agents, combining classical parasitological techniques with molecular diagnostics, with a particular focus on *E. granulosus*, *E. multilocularis*, canine parvovirus, and canine coronavirus.

## Materials and methods

### Study area

The study area includes the Foreste Casentinesi, Monte Falterona e Campigna National Park (FCNP), which encompasses 368 km$^2$ of protected areas across Tuscany and Emilia-Romagna regions, spanning the provinces of Arezzo, Florence, and Forlì-Cesena. Altitudes ranges from 400 to 1658 m a.s.l. and most of the territory is covered by forests, with typical temperate-sub-Mediterranean vegetation.

 

The FCNP covers a spectrum of environments, from pristine natural areas with little to no human impact to urban centers where traditional agriculture, including native livestock breeding, historic landmarks, and tourist destinations are preserved. Despite this, the park remains sparsely populated, with only 2,000 inhabitants across its entire protected area. This creates an ideal environment for wildlife, as this area is densely populated by large mammals – red deer (*Cervus elaphus* L., 1758), fallow deer (*Dama dama* L., 1758), roe deer (*Capreolus capreolus* L., 1758), wild boar (*Sus scrofa* L., 1758), mouflon (*Ovis musimon* Pallas, 1762), and wolf (*Canis lupus italicus* A., 1921), along with a diversity of species both vertebrate and invertebrate (www.parcoforestecasentinesi.it, accessed on 26 November 2024).

## Sample collection

Wolf feces were collected as part of a non-invasive genetic sampling conducted within the FCNP from May 2019 to March 2020, as described in Dissegna et al. (2023) [27]. The FCNP was divided into 30 cells using a standardized 5x5 km grid, with at least one transect per cell. A total of 39 transects, each 5 km in length, were surveyed twice per month for four months, resulting in a total sampling effort of approximately 1,600 km. Geolocation data was recorded for each sample. Samples were collected using a fecal-swab protocol for genetic analysis, followed by the collection of the entire stool. Genetic analysis was carried out at the Italian Institute for Environmental Protection and Research (ISPRA) laboratory to determine the host species and individual identity of wolves. Details regarding the population monitoring are available in Dissegna et al. (2023) [27]. Fecal samples were transported to the Belluno Laboratory of the Istituto Zooprofilattico Sperimentale delle Venezie (IZSVe) for virological and parasitological analyses. For safety reasons, the samples were subjected to a temperature of −80°C for 10 days [28], and then stored at −20°C until further analysis.

## Virological analysis

Virological analyses were conducted only on samples corresponding to unique genotypes (i.e., unique individuals) to assess the circulation of parvovirus and coronavirus. When more than one sample corresponded to the same individual, the latest collected sample was used. CPV and CCoV were selected as target pathogens because these viruses are highly contagious, environmentally stable, and can persist in fecal material, making them ideal candidates for non-invasive detection.

Samples were clarified with antibiotic-treated Phosphate-Buffered Saline (PBSA) and nucleic acids were extracted using the QIAamp®Viral RNA Mini kit (QIAGEN, 2020). We screened samples for the presence of coronaviruses using a broad-spectrum nested PCR able to detect members of all known genera, as detailed elsewhere [29]. For the identification of CPV-2, a real-time PCR was performed using the QuantiFast® Pathogen PCR+IC amplification kit (QIAGEN, Hilden, Germany) implemented in the CFX 96 BIO-RAD (BIO-RAD, Hercules, CA, USA), as detailed elsewhere [30].

Positive samples were further characterized through Sanger sequencing. In particular, a partial fragment of the RNA-dependent RNA polymerase (RdRp) gene from a nested PCR-positive sample was sequenced. The sequence was submitted to GenBank under the accession number PV661850.

A phylogenetic analysis was performed to compare the virus identified in the wolf with similar sequences retrieved from GenBank, with particular focus on the closest strains and strains that were previously found in dogs and wild carnivores from Italy. A maximum likelihood phylogenetic tree was constructed using PhyML (version 3.0), implemented in Seaview (Lyon, France), with a GTR+G4 substitution model. The tree was visualized using the online software iTol (https://itol.embl.de/) to highlight the relationship with deposited strains identified in various domestic and wild carnivores.

## Parasitological analysis

All collected fecal samples were examined with classical copromicroscopy (sedimentation-flotation technique), using a solution composed of sucrose, sodium nitrate and water (specific gravity: 1.300) to detect helminth eggs and protozoan (oo)cysts. Parasitic elements were identified at genus or species level based on their morphology [31,32].

All samples attributable to individual host genotypes, as well as fecal samples that tested positive for taeniid eggs at copromicroscopy (whenever sufficient material was available) were analyzed using a real-time duplex PCR to detect *Echinococcus granulosus* and *E. multilocularis*. For DNA extraction, 0.10 g of feces was aliquoted into 2 ml Eppendorf tubes. A 5 mm bead and 1 ml of PBSA were added to each tube. The feces were then homogenized using the Tissue Lyser (QIAGEN) at 30 Hz for 5 minutes and centrifuged at 1000 g for 1 minute. Finally, 200 µl of the sample and 9 µl of internal control (QuantiNova Internal Control DNA – QIAGEN) were extracted using the Hamilton Microlab® STAR automated extractor (HAMILTON), following the protocol of the MagMAX™ Viral/Pathogen Nucleic Acid Isolation Kit (QIAGEN). The extracted DNA was amplified using a real-time duplex assay for the identification of *E. granulosus* [33] and *E. multilocularis* [34], employing the QuantiNova® Pathogen+IC Kit (QIAGEN).

### Data analysis

Fecal samples could not be treated as independent, as multiple samples were collected from the same individuals. Therefore, we calculated the frequency, or positivity rate, as the proportion of positive samples out of the total number of samples analyzed for the pathogen under consideration.

To calculate prevalence, we considered only the freshest samples of the genetically identified individuals. Prevalence values and their 95% confidence intervals (95% CI) were calculated using the Wilson method in EpiTools (www.epitools.ausvet.com.au) for each identified pathogen.

The potential effects of spatial (north *vs* south of the park) and temporal (warm *vs* cold season) on the presence or absence of parasites in all samples (based on geographical coordinates and date of sampling) were assessed using Pearson's Chi-square test or Fisher's exact test. The influence of individual factor (i.e., sex) was similarly investigated, considering only samples from genetically identified individuals. Only parasitic taxa with prevalence values higher than 10% were investigated with separate tests. All analyses were carried out in IBM SPSS Statistics 29.0 and P value <0.05 were considered statistically significant.

To create maps illustrating the spatial distribution of the samples, the park's boundaries were downloaded from the institutional Web GIS (https://www.parcoforestecasentinesi.it/it/multimedia/webgis). The spatial data were then plotted using the GIS software ArcGIS Pro 3.4.2.

### Ethics statement

This study did not involve any live animal handling. All samples were collected non-invasively from the environment; therefore, no ethical approval was required. Fieldwork procedures were specifically approved by ISPRA as a part of national wolf monitoring activities.

### Results

All relevant data are reported in the sections below, with complete datasets available in S1 Table. A total of 261 scat samples were transported to the Belluno Laboratory, IZSVe. Of these, 132 samples (50.6%) were successfully genotyped, identifying 81 unique genotypes, comprising 80 wolves (44 females, 34 males, and 2 undetermined) and 1 dog. The dog sample was excluded from subsequent analyses. From the population monitoring, eight packs were confirmed within the FCNP (Fig 1). Further details on the population monitoring results are available in Dissegna et al. (2023) [27].

### Virological analysis

None of the 80 tested samples were positive for parvovirus, while only one sample, from a she-wolf of pack H, resulted positive for coronavirus (1.2%; IC 95%: 0.2–6.7). The partial RdRp gene sequence (approximately 400 base pairs) obtained through the Sanger sequencing approach from the positive amplicon allowed the putative classification of this

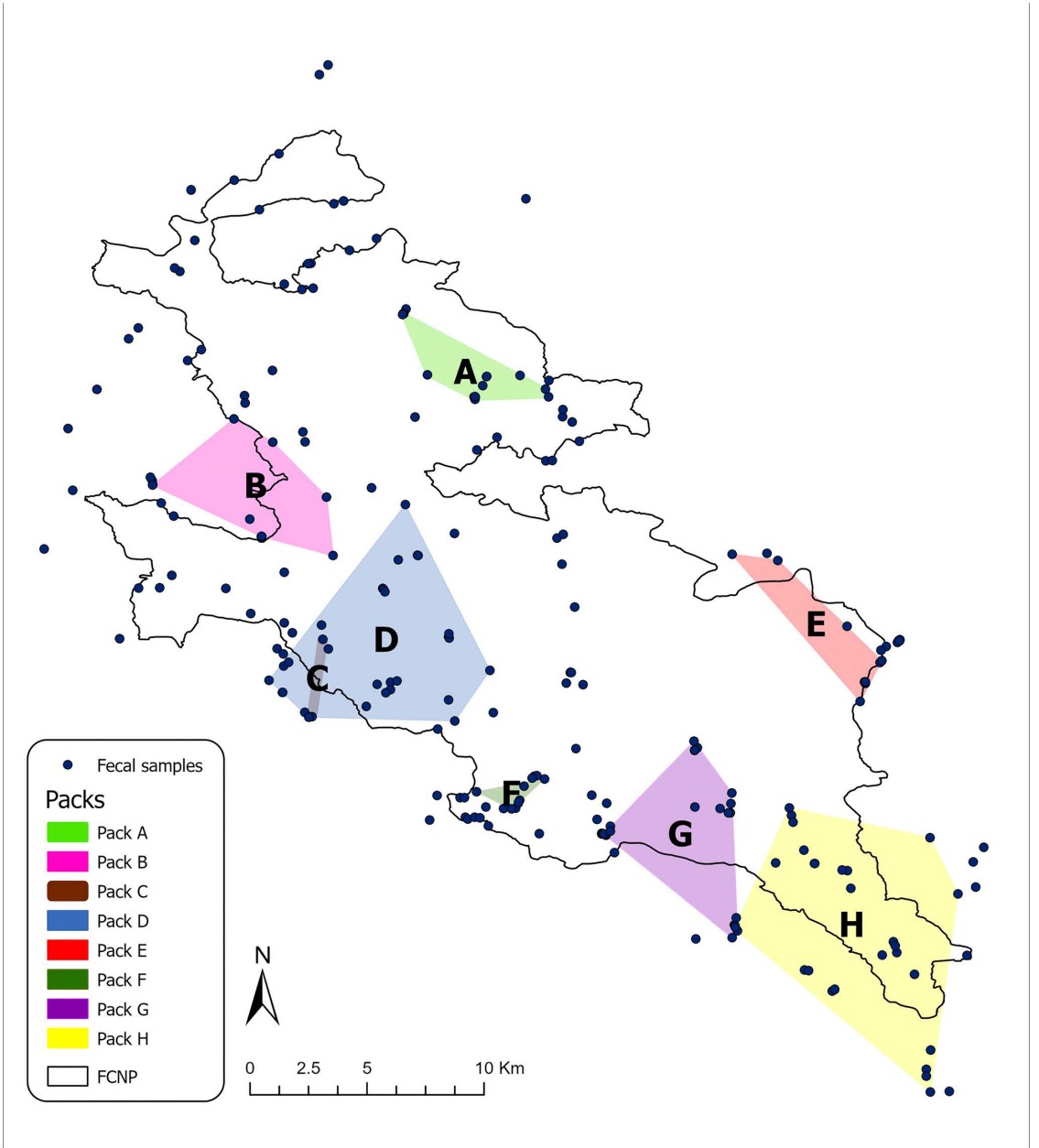

**Fig 1. Distribution of wolves' samples and identified packs.** This image shows the distribution of wolves' fecal samples and identified packs (modified from Dissegna et al., 2023). For further details on population monitoring, see Dissegna et al., 2023 [27].

strain as CCoV under the species *Alphacoronavirus 1.* Our sequence clustered with others previously identified in wolves and foxes in Italy. It displayed the highest nucleotide and amino acid identity (98.3% and 100%, respectively) with a CCoV strain identified in the same host species in Abruzzo (GenBank accession number OR042794). This cluster also included a single strain isolated from a dog in Apulia, which shared 93.41% nucleotide and 97.7% amino acid identity with our sequence (Fig 2).

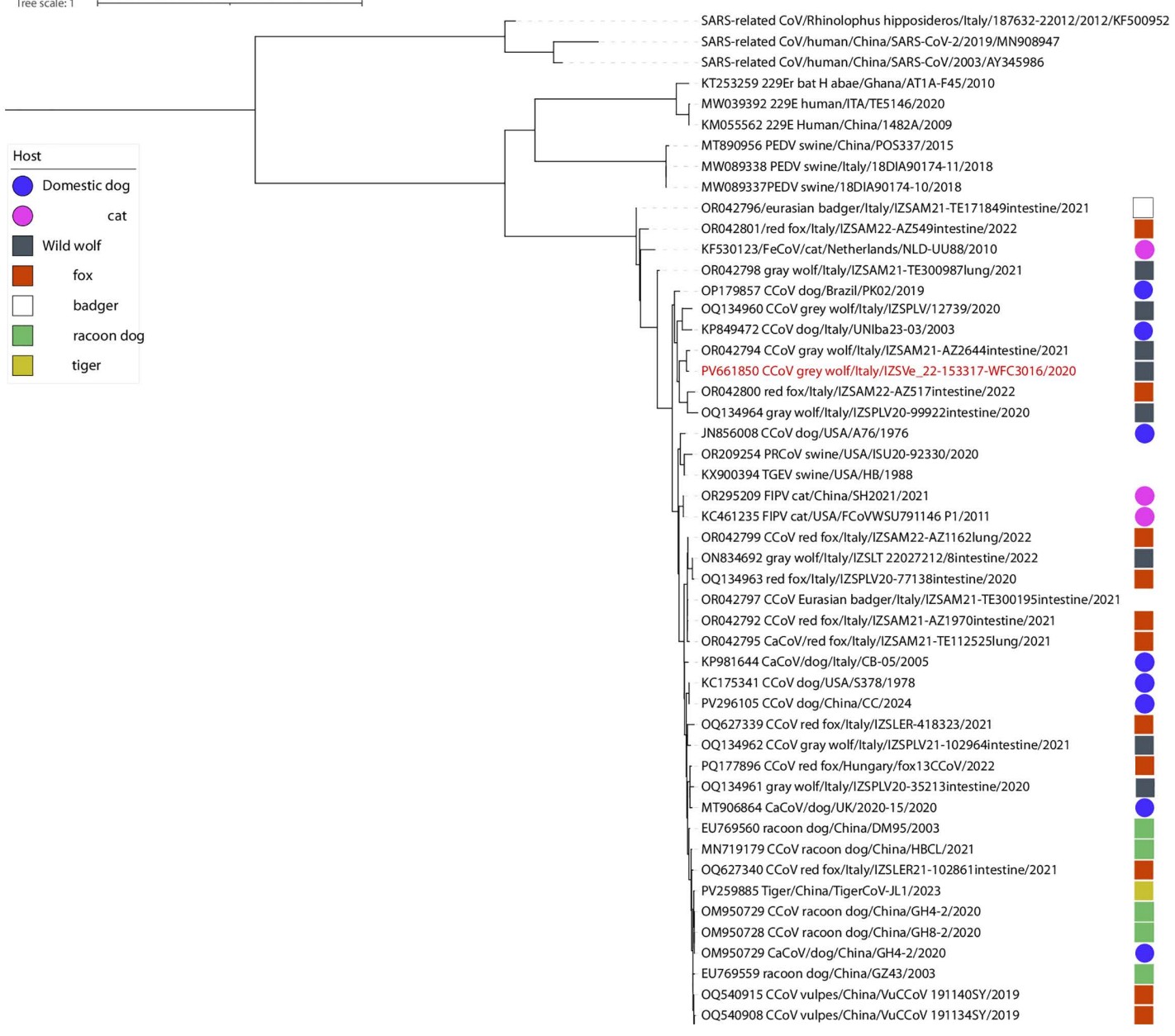

**Fig 2. Maximum Likelihood (ML) phylogenetic tree based on partial *RdRp* gene sequences.** The original sequence obtained in this study is highlighted in red. Circles and squares represent sequences identified in domestic and wild carnivores, respectively, with colors corresponding to host species as indicated in the legend. Specifically, sequences from dogs are shown in blue circles, while those from wolves are shown in grey squares.

## Parasitological analysis

Overall, 260 fecal samples were analysed through copromicroscopy, with 96.1% (95% CI: 93.1–97.9) testing positive for at least one parasite species. A complete list of the detected parasites is presented in Table 1, along with the proportion of positive samples (frequency) and proportion of positive individuals (prevalence).

**Table 1. Frequency and prevalence values of the identified parasites through copromicroscopy examination.**

| Taxonomic group | Number of positive samples (Frequency %)<br>$n=260$ | Number of positive individuals (Prevalence %)<br>$n=80$ | Confidence interval (95%) |
|---|---|---|---|
| *Eucoleus* spp. | 232 (89.2) | 72 (90.0) | 81.5–94.8 |
| *Sarcocystis* spp. | 109 (41.9) | 34 (42.5) | 32.3–53.4 |
| Ancylostomatidae | 73 (28.0) | 21 (26.2) | 17.9–36.8 |
| Taeniidae | 61 (23.5) | 23 (28.7) | 20.0–39.4 |
| *Trichuris vulpis* | 7 (2.7) | 1 (1.2) | 0.2–6.7 |
| *Toxocara canis* | 5 (1.9) | 1 (1.2) | 0.2–6.7 |
| Coccidia | 3 (1.1) | 1 (1.2) | 0.2–6.7 |

The table shows the number of positive samples and the number of positive individuals. The proportion of positive samples is reported as the frequency, while the proportion of positive individuals is reported as the prevalence. 95% confidence intervals are also presented.

Notably, 176 (67.7%) of the analyzed samples exhibited co-infestations, with up to five parasite species detected in a single sample. Specifically, 121 (46.5%) samples tested positive for two species, 47 (18.0%) for three species and 7 (2.7%) for four species. *Eucoleus* spp. and the protozoan *Sarcocystis* spp. were the most prevalent parasites, each with a frequency exceeding 40%. Ancylostomatidae and Taeniidae were also relatively common, with prevalence rates above 20%. All other detected parasites had a prevalence lower than 2%. Details of the parasites detected in each wolf pack are shown in Fig 3.

In consideration of their prevalence value, *Eucoleus* spp., Ancylostomatids, Taeniids, and *Sarcocystis* spp. data were analysed to investigate the influence of individual (sex), spatial and temporal factors (north *vs* south of the park; warm *vs* cold season), but no significant differences were encountered ($P>0.05$).

All samples corresponding to individual host genotypes ($n=80$), together with fecal samples found positive for taeniid eggs at copromicroscopy (provided that sufficient material was available, $n=24$), were subjected to real-time duplex PCR for the detection of *E. granulosus* and *E. multilocularis*. In total, 104 samples were molecularly analyzed, and none tested positive.

## Discussion

This study assessed the helminth fauna and the circulation of parvovirus and coronavirus in the wolf population of the FCNP, contributing to a better understanding of pathogen circulation in wolves and potentially informing future conservation and management strategies for the species.

Infectious diseases can play a significant role in shaping wildlife population dynamics, and among those affecting wolves, Carnivore Protoparvovirus 1 (CPPV-1) and Canine Enteric Coronavirus (CCoV) are of particular concern. In our study, only one positive case to coronavirus (prevalence of 1.2%) was detected in a female wolf belonging to pack H, suggesting either low prevalence within the pack or a sporadic spillover case from dogs. This data is consistent with previous studies in Italy, where the virus was either not detected at all or found at low prevalence levels (<10%). In the Majella National Park (MNP) none out of two packs ($n=20$ fecal samples) tested positive for CCoV [35]. Moreover, negative molecular findings on tissue samples were described from Emilia-Romagna, Tuscany, and Calabria [36], while negative serology was confirmed in 1996 ($n=4$ wolves) [37]. However, it is important to mention how sample size in this study could have hampered identification of the virus if circulating at low prevalence, such as the one identified in our territory. In Italy, the highest prevalence of 8.6% was identified through a non-invasive study conducted on fecal samples in wolves from the Abruzzo Lazio Molise National Park (ALMNP). As highlighted by the authors, in the park a population of unvaccinated free-ranging dogs coexists with wolf packs, which could have facilitated dog-to-wolf transmission [15].

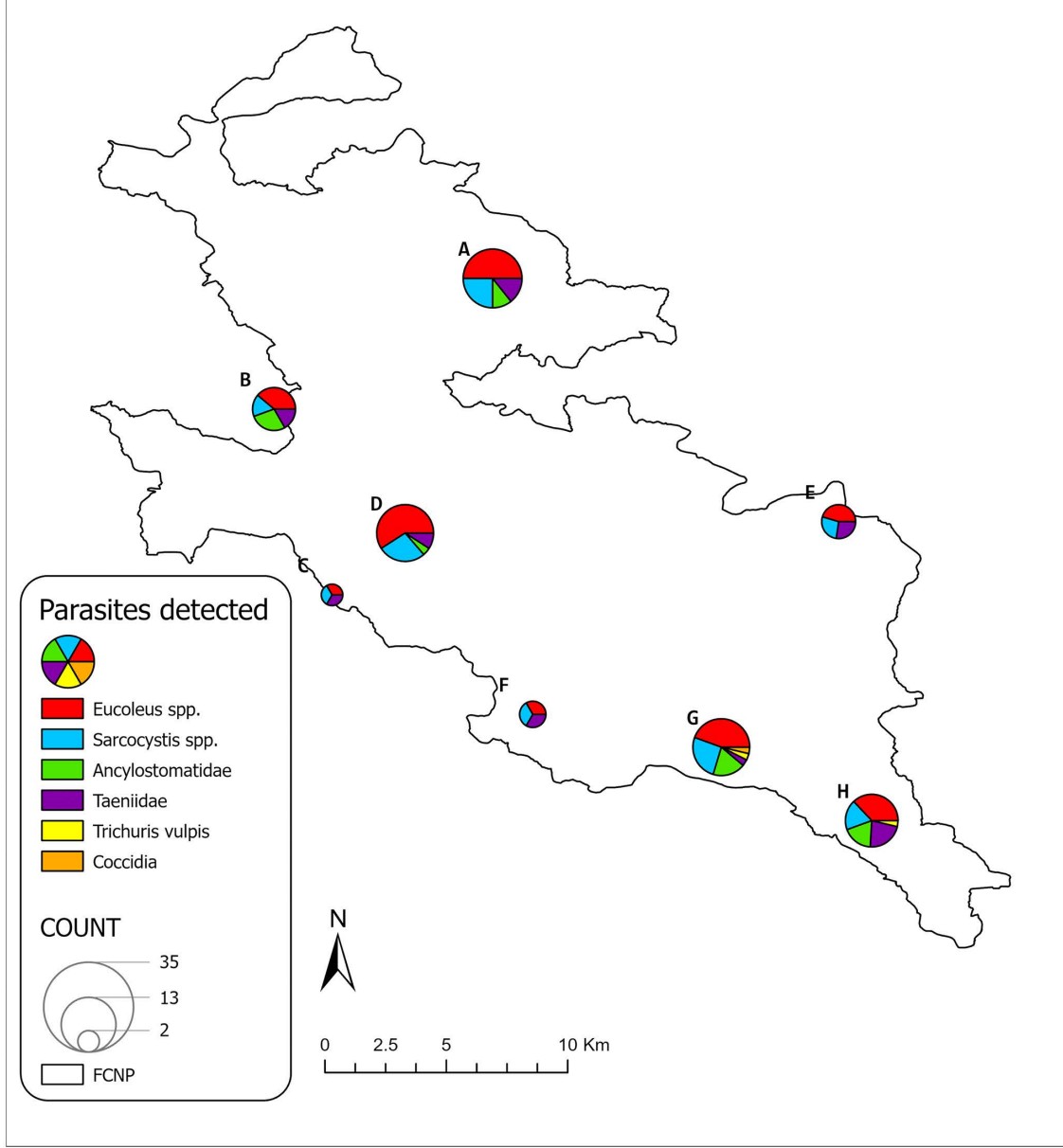

**Fig 3. Parasites detected in each wolf pack.** The image displays the frequency of parasites detected in each wolf pack, as identified by Dissegna et al., 2023 [27]. The size of the pie charts is proportional to the number of samples collected per pack.

Low detection rate in wolves from our study could be related with the occurrence of spillover events from dogs. Although available phylogenetic data tend to support the clustering of strains within wildlife species rather than interspersed between dogs and wolves, it is crucial to highlight how molecular data are still scarce, especially in dogs. Most existing data are related with partial sequences of *RdRp*, which offer low phylogenetic resolution. To better assess the dynamics at the domestic-wildlife interface, we recommend enhanced surveillance in domestic dog populations and the use of whole genome sequencing of local viral strains. This approach would allow to more accurately distinguish between

the hypotheses of recurrent spillover versus sustained transmission within wildlife, trace infection sources, and evaluate potential risks to wild populations.

No positive case to parvovirus was found in our study, similar to previous non-invasive sampling of feces performed in the same area and accounting for similar sample size [38]. However, previous cases of CPV-2 infection were reported in the park from dead animals. For example, CPV-2 infection was confirmed in 2019 from two wolf pups found dead, one of which had been hit by a vehicle. To determine the cause of death, necropsies were performed, revealing lesions associated with CPV infection, which was subsequently confirmed by molecular analysis, presuming CPV-2 as cause of death or severe illness [39]. The virus was also previously identified in FCNP in the tissues of dead wolves between 2005 and 2007 with signs of trauma from road accidents [38]. These data support a low sensitivity of opportunistic fecal sampling for the detection of CPV-2 most likely because this virus has highest prevalence and shedding in pups rather than a homogeneous distribution among the pack. In similar cases, passive surveillance on carcasses and serological screening are a better tool to monitor viral circulation, as shown in the literature [6,11,14,36,38]. Specifically, Musto (2021) [36] identified viral DNA in tissue samples with detection rates of 45.4% in Emilia-Romagna, 29.5% in Tuscany, and 25.0% in Calabria. Based on fecal material, a prevalence of 15.2% for CPV-2 was recorded in the ALMNP in 2006–2007 [15]. It is widely recognized that domestic dogs may still represent the main reservoir for the virus even in countries with high dog vaccination rates, such as Italy. In these countries, legal and illegal trade of dogs and other carnivores is likely responsible for recurrent introduction of novel strains and could be a main risk to wildlife health [11,40].

From a public health perspective, Taeniidae is the most relevant parasitic family in wild canids, as it includes *Echinococcus* spp. and certain *Taenia* species. Wolves serve as definitive hosts, thereby contributing to the environmental dissemination of these parasites. In our survey, taeniid eggs were detected with a prevalence of 28.7%. Considering other studies conducted in FCNP, our results showed a considerably higher value than the 7.2% reported by Ambrogi et al. (2019) [38] but lower than the 61.1% recorded by Poglayen et al. (2017) [21]. However, our result may be underestimated due to the low sensitivity of copromicroscopy in detecting taeniid eggs and the intermittent shedding of eggs in feces [41]. Following the method proposed by Obber et al. (2022) [42] to increase the sensitivity of *E. granulosus* and *E. multilocularis* detection at the expected low prevalence, we performed a real-time duplex PCR on samples attributable to individual host genotypes – regardless of their copromicroscopic results – as well as on those that tested positive for taeniid eggs at copromicroscopy. This approach aimed to roughly assess the prevalence (i.e., the percentage of positives among genotyped samples) and to increase the likelihood of detecting the parasite if circulating, including in non-genotyped samples. Consistent with the findings of Poglayen et al. (2017) [21], no case of *E. multilocularis* was detected. In contrast, while Poglayen et al. reported a low prevalence of *E. granulosus* (G1-G3) (5.5%), we did not detect any cases. At least two decades have passed since the collection of samples analyzed in Poglayen's study, and our findings reaffirm that wolves in this national park do not pose a significant risk for this zoonotic parasite, nor does a sylvatic cycle appear to be present. The previously low prevalence of *E. granulosus* (G1-G3) in wolves may have been linked to predation on domestic animals; therefore, our results could indicate that the park's management measures have effectively minimized the already low incidence of wolf predation on livestock. Consequently, the wolf diet in this park is now primarily based on wild ungulates, which seem to be free from *Echinococcus* infection. We did not perform molecular analysis to further characterize the taeniids eggs. In the study conducted by Poglayen et al. (2017), the presence of *Taenia hydatigena* (40.7%), *T. krabbei* (22.2%), *T. polyacantha* (1.8%), along with *Echinococcus granulosus* (5.5%), were reported. Given their indirect life cycle, the occurrence of these tapeworms is closely linked to the diet of wolves. Except for *T. hydatigena*, which is associated with both domestic and wild ungulates, all other taeniid species previously identified are part of a sylvatic cycle [43,44]. This further suggests that wolves in this area primarily prey on wild animals. Continuous monitoring of wolves is desirable to better understand the ecological dynamics of the species and its parasites in an era of social and climatic change. This need is further supported by recent evidence from Cafiero et al. (2025) [45], who reported the detection of wolves positive for the *E. multilocularis* genome in northern Tuscany.

Among helminth species of veterinary relevance, the most frequently detected genus in our survey (prevalence of 90.0%) was *Eucoleus* spp. (synonym: *Capillaria* spp.), a group of cosmopolitan nematodes primarily affecting canids. *Eucoleus aerophilus* (syn. *Capillaria aerophila*) and *Eucoleus boehmi* (syn. *Capillaria boehmi*) are the bronchial and nasal capillariids, respectively, whose eggs are found in the host's feces. The life cycle of these nematodes remains unclear, with speculation that earthworms may serve as intermediate or paratenic hosts, but direct ingestion of infective third stage larvae in eggs cannot be excluded [46,47]. The capillariids eggs were degraded after being stored at −80°C, preventing the distinction between the two species, *E. boehmi* and *E. aerophilus* [46]. Interestingly, this genus is not typically considered the most prevalent parasite in wolves [43]. Such high prevalence values have not been reported in other European studies outside Italy [48,49], as the highest recorded prevalence in Europe was 30% in captive wolves in Germany [50]. In Italy, prevalence rates in wolves vary across regions, ranging from 80% for *E. boehmi* and 20% for *E. aerophilus* in the ALMNP [51], 67% and 32% for the same species in northern Tuscany [52], to 35% for *E. aerophilus* in the MNP [35], and as low as 2.3% for *Eucoleus* spp. in Piedmont [53]. A recent study comparing nematodes in wolves and foxes across both anthropized (Pisa province, Italy) and natural environments (Monterufoli Caselli Nature Reserve, Pisa province, Italy, and FCNP) found that capillariids were significantly more prevalent in wolves (~60%) than in foxes (~30%) in both settings [54]. These findings highlight the potential role of wolves in the transmission cycle of these parasites, as the red fox has traditionally been considered the primary reservoir in Europe. Given these results, wolves may contribute significantly to environmental contamination with capillariid eggs, posing a risk of infection to other animals, including domestic dogs and cats. Additionally, *E. aerophilus* presents a zoonotic risk, as it is the causative agent of human pulmonary capillariasis [55].

Ancylostomatidae was detected with a prevalence of 26.2%. However, this may be underestimated, as their eggs are less resistant to unfavorable conditions, such as freezing −80°C and thawing. Ancylostomatidae are among the most commonly detected parasites in wolves, with reported prevalence rates exceeding 60% [43]. *Uncinaria stenocephala* is the more prevalent species in Italy, with reported prevalence rates of 67% compared to 16% for *Ancylostoma caninum* [56] and 26.2% *versus* 7.1% in Piedmont [53]. More recently, Perrucci et al. (2023) [54] found a hookworm prevalence of 53.8% in wolf samples from natural areas, suggesting a higher prevalence in wolves from these environments compared to those from anthropized areas. This difference is likely due to more favorable conditions for egg development, such as higher humidity and shade, which facilitate direct transmission.

In our survey, *Trichuris vulpis* and *Toxocara canis* were detected sporadically, each with a prevalence of 1.2%. The prevalence of *T. vulpis* is consistent with values reported in other wild canids in Italy, such as foxes [57,58]. In contrast, the prevalence of *T. canis* may be underestimated due to sampling bias towards adult wolves, as fecal samples collected along transects are predominantly associated with adult scent-marking behavior. Indeed, ascarid infections are generally more common in juveniles, owing to vertical transmission routes [59]. Additionally, the role of paratenic hosts is central to the epidemiology of *T. canis*, with much higher prevalence reported in canids that prey on small vertebrates [54,57,60,61].

Among protozoans, we detected *Sarcocystis* spp. with a high prevalence of 42.5%. *Sarcocystis* spp. is an apicomplexan protozoan with a cycle that typically involves a predator-prey relationship, in which herbivores (or omnivores) serve as intermediate hosts, developing cysts within the muscle tissue (sarcocysts), and carnivores (or omnivores) as definitive hosts, excreting sporocysts in their feces, which may contaminate food or water [62]. Consequently, the high prevalence of *Sarcocystis* spp. detected in our study could be strongly associated with the diet of wolves in the FCNP. It has been observed that when wolves primarily feed on wild ungulates, the *Sarcocystis* spp. load increases, whereas a more varied diet helps reducing parasite burden [51]. In Germany, where wolves predominantly prey on fallow deer and wild boar, prevalence rates reach as high as 95%. Molecular analyses have identified 12 potential *Sarcocystis* species, with the most prevalent (*S. taeniata, S. gracilis, S. capreolicanis, S. grueneri, S. tenella, S. miescheriana,* and *S. cruzi*) being linked to the intermediate hosts present in Germany [63]. The presence of *Sarcocystis* spp. in wolves in Italy remains poorly studied. Indeed, molecular analyses are necessary for the specific identification of *Sarcocystis*, which would allow a better understanding of both the predator–prey relationship and potential public health risks. On the contrary, research

on intermediate hosts is more extensive. A high prevalence of *S. miescheriana* and a sporadically presence of *S. suihominis* is well documented in wild boars [64–68]. Molecular analyses have also identified multiple *Sarcocystis* species in roe deer (e.g., *S. gracilis, S. silva, S. capreolicanis,* and *S. linearis*) [69]. Based on our findings, it is likely that this overlooked parasitosis is also present in ungulates within the FCNP, and its potential public health implications should not be underestimated.

Coccidia were detected at low prevalence (1.2%) in this study, in contrast to a previous survey conducted in the same area [38]. Other studies suggest that wild canids may play a role in the environmental maintenance and transmission of these parasites to both wildlife and domestic animals [54]. The pathogenic impact in wolves, particularly juveniles, remains unclear and warrants further investigation, given documented cases of coccidia-associated mortality [70].

In conclusion, we investigated the health status of the wolf population in the FCNP. We found a low prevalence of coronavirus shedding among these wolves and the detected sequence clustered with others from wild canids from Italy. However, molecular data on domestic dogs remains limited. This highlights the need of surveillance in this species, to more accurately monitor pathogen transmission within wildlife and domestic animals, trace infections, and assess potential conservation risks. From a public health perspective, our results indicate that wolves in the FCNP do not represent a significant risk for the transmission of zoonotic parasites, such as *Echinococcus* spp. Nevertheless, the role of wolves in the epidemiology of other parasites of veterinary relevance must be underlined. Copromicroscopic analysis proved to be a valuable, non-invasive tool for monitoring this wild population, providing useful insights into the main parasite genera circulating among these wolves. However, its low sensitivity toward certain parasite groups (e.g., taeniids) should be considered when interpreting the results. Nonetheless, integrating molecular methods allows for a more precise identification of the pathogens involved. Combining necroscopic analysis of deceased wolves with copromicroscopic examination of environmental fecal samples would significantly enhance monitoring efforts. This approach is particularly valuable because the fecal shedding of certain pathogens occurs only during specific stages of infection, potentially leading to false negatives if relying solely on fecal testing. By employing both methods, it is possible to detect fecal-shed parasites at a relatively low cost using copromicroscopic analysis, while necropsy can provide crucial information on exposure to pathogens not consistently detectable in feces, such as parvoviruses or coronaviruses. An integrated approach offers a more comprehensive understanding of the health status of wild canids, with important implications for both wildlife conservation and public health.

## Supporting information

**S1 Table. Complete dataset of fecal samples analyzed, including sample ID, collection place and date, genotyped sample, wolf sex, and virological and parasitological results.**
(XLSX)

## Author contributions

**Conceptualization:** Federica Obber.

**Data curation:** Elisabetta Ferraro, Graziana Da Rold, Roberto Celva, Federica Obber.

**Formal analysis:** Elisabetta Ferraro, Stefania Leopardi, Rudi Cassini.

**Funding acquisition:** Nadia Cappai, Carlo Vittorio Citterio.

**Investigation:** Elisabetta Ferraro, Graziana Da Rold, Roberto Celva, Elisa Dalla Libera, Paola De Benedictis, Arianna Dissegna, Federica Obber.

**Methodology:** Graziana Da Rold, Stefania Leopardi, Arianna Dissegna.

**Project administration:** Nadia Cappai.

**Resources:** Graziana Da Rold, Nadia Cappai.

**Supervision:** Giulia Simonato, Carlo Vittorio Citterio, Rudi Cassini, Federica Obber.

**Visualization:** Elisabetta Ferraro, Stefania Leopardi, Federica Obber.

**Writing – original draft:** Elisabetta Ferraro.

**Writing – review & editing:** Elisabetta Ferraro, Graziana Da Rold, Roberto Celva, Elisa Dalla Libera, Stefania Leopardi, Giulia Simonato, Paola De Benedictis, Nadia Cappai, Arianna Dissegna, Carlo Vittorio Citterio, Rudi Cassini, Federica Obber.

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
