## [Decision Letter · Decision Letter 0]

6 Oct 2025

Dear Dr. Ferraro,

Thank you for submitting your manuscript to PLOS ONE. After careful consideration, we feel that it has merit but does not fully meet PLOS ONE’s publication criteria as it currently stands. Therefore, we invite you to submit a revised version of the manuscript that addresses the points raised during the review process.

Both reviewers agree that your study is important, but there are mistakes in the use of scientific designations. Also, test methods need further clarification.

We look forward to receiving your revised manuscript.

Kind regards,

Ulrike Gertrud Munderloh, Ph.D.

Academic Editor

PLOS ONE

Journal Requirements:

3. We note that Figures 1 & 3 in your submission contain [map/satellite] images which may be copyrighted. All PLOS content is published under the Creative Commons Attribution License (CC BY 4.0), which means that the manuscript, images, and Supporting Information files will be freely available online, and any third party is permitted to access, download, copy, distribute, and use these materials in any way, even commercially, with proper attribution. For these reasons, we cannot publish previously copyrighted maps or satellite images created using proprietary data, such as Google software (Google Maps, Street View, and Earth). For more information, see our copyright guidelines: http://journals.plos.org/plosone/s/licenses-and-copyright.

a. You may seek permission from the original copyright holder of Figures 1 & 3 to publish the content specifically under the CC BY 4.0 license.

Please upload the completed Content Permission Form or other proof of granted permissions as an "Other" file

Reviewers' comments:

Reviewer's Responses to Questions

**Comments to the Author**

1. Is the manuscript technically sound, and do the data support the conclusions?

Reviewer #1: Yes

Reviewer #2: Yes

2. Has the statistical analysis been performed appropriately and rigorously?

Reviewer #1: Yes

Reviewer #2: Yes

3. Have the authors made all data underlying the findings in their manuscript fully available?

Reviewer #1: Yes

Reviewer #2: Yes

4. Is the manuscript presented in an intelligible fashion and written in standard English?

Reviewer #1: Yes

Reviewer #2: Yes

Reviewer #1: Dear Authors,

This is a very interesting study reporting data on some viruse and endoparasite infections of wolves in an Italian national park, by examining a fairly large number of wolf fecal samples collected over almost a year (between May 2019 and March 2020). Overall, the manuscript is extremely well conceived and written, and the results are really very interesting. However, the manuscript shows some small imperfections in the way the names of the studied pathogens were written, while the discussion on the sensitivity of the method used for the detection of Echinococcus granulosus and Echinococcus multilocularis eggs deserves further exploration. More detailed comments are reported below.

Line 26: Sarcocystis spp. are not helminths

Line 27: Please, replace Coccidia with coccidia

Line 38: Please, replace extirpated with a more appropriate term

Lines 56 and 59: Carnivore Protoparvovirus 1, Canine Enteric Coronavirus and Canine Parvovirus 2 should not be written in italics

Lines 153-154: The coproscopic method used is not highly sensitive for the detection of Taeniid eggs. For this reason, other available methods are more frequently used, see Mathis A, Deplazes P, Eckert J. An improved test system for PCR-based specific detection of Echinococcus multilocularis eggs. Journal of Helminthology. 1996;70(3):219-222. doi:10.1017/S0022149X00015443. This needs to be discussed.

Line 155: Please, replace cysts with (oo)cysts

Line 280: "Taeniidae is the most relevant parasitic genus"; Taeniidae is a parasitic family and not a parasitic genus

Lines 284-288: Having used a technique that is not highly sensitive for the detection of Taeniids, even the use of PCR can give false results if performed only on samples that tested positive to copromicroscopy

Line 296: Please, replace taeniids with taeniid

Lines 280-359: In the discussion section, parasitological data should also be compared with those obtained in the following recent studies:

1. Cafiero SA, Petroni L, Natucci L, Tomassini O, Romig T, Wassermann M, Rossi C, Hauffe HC, Casulli A, Massolo A. New evidence from the northern Apennines, Italy, suggests a southward expansion of Echinococcus multilocularis range in Europe. Sci Rep. 2025 Mar 1;15(1):7353. doi: 10.1038/s41598-025-91596-7

2. Minichino A, Ciuca L, Dipineto L, Rinaldi L, Montagnaro S, Borrelli L, Fioretti A, De Luca Bossa LM, Garella G, Ferrara G. Exposure to selected pathogens in wild mammals from a rescue and rehabilitation center in southern Italy. One Health. 2025 Apr 21;20:101049. doi: 10.1016/j.onehlt.2025.101049

3. Cafiero SA, Petroni L, Natucci L, Casale L, Raffaelli M, Baldacci D, Di Rosso A, Rossi C, Casulli A, Massolo A, Hauffe HC, Perrucci S. Parasite diversity in grey wolves (Canis lupus) from Tuscany, central Italy: a copromicroscopical investigation. Int J Parasitol Parasites Wildl. 2025 May 31;27:101092. doi: 10.1016/j.ijppaw.2025.101092

Lines 368-369: This sentence should be deleted or changed according to comments reported above

Line 380: Please, replace Reference with References

Lines 381-553:In the references, parasite and animal genera and species should be written in italics

Reviewer #2: Line 43 - What type of legal protection?

Lines 50-51 - Include an example of a zoonotic disease that corresponds to this assertion.

Line 82 - need scientific name of red fox

Line 88 - Echinococcus granulosus sense strict includes G1-G3 but just G1 is mentioned here, what about G2 and G3?

Line 96 - wolf should be wolves

Line 158-159 - which unique genotypes are tested for?

Line 287 - again which genotypes were test for?

Generally an interesting and well written paper. Are Parvo and Coronavirus really the only two viruses of concern between feral/domestic dogs and wolves? Perhaps dive a little more into why these two were chosen and their history in the area, in contrast to other transmissible viruses of concern.

what does this mean?). If published, this will include your full peer review and any attached files.

**Do you want your identity to be public for this peer review?** For information about this choice, including consent withdrawal, please see our Privacy Policy

Reviewer #1: No

Reviewer #2: No

While revising your submission, please upload your figure files to the Preflight Analysis and Conversion Engine (PACE) digital diagnostic tool, https://pacev2.apexcovantage.com/. PACE helps ensure that figures meet PLOS requirements. To use PACE, you must first register as a user. Registration is free. Then, login and navigate to the UPLOAD tab, where you will find detailed instructions on how to use the tool. If you encounter any issues or have any questions when using PACE, please email PLOS at figures@plos.org

---

## [Author Response · Author response to Decision Letter 1]

18 Nov 2025

Dear Editor,

We thank the Reviewers for their valuable comments and constructive suggestions, which have helped us to improve the quality and clarity of our manuscript. Below, we provide a detailed response to each comment.

Reviewer #1

Dear Authors,

This is a very interesting study reporting data on some viruses and endoparasite infections of wolves in an Italian national park, by examining a fairly large number of wolf fecal samples collected over almost a year (between May 2019 and March 2020). Overall, the manuscript is extremely well conceived and written, and the results are really very interesting. However, the manuscript shows some small imperfections in the way the names of the studied pathogens were written, while the discussion on the sensitivity of the method used for the detection of Echinococcus granulosus and Echinococcus multilocularis eggs deserves further exploration. More detailed comments are reported below.

Reply: We thank the Reviewer for comments and suggestions. The manuscript was revised accordingly, as specified below.

R#1 Line 26: Sarcocystis spp. are not helminths

Reply: We replaced helminths with a more appropriate term, that is “endoparasites”.

R#1 Line 27: Please, replace Coccidia with coccidia

Reply: We replaced coccidia as suggested.

R#1 Line 38: Please, replace extirpated with a more appropriate term

Reply: We changed with “driven to near extinction”.

R#1 Lines 56 and 59: Carnivore Protoparvovirus 1, Canine Enteric Coronavirus and Canine Parvovirus 2 should not be written in italics

Reply: We corrected the italics as requested.

R#1 Lines 153-154: The coproscopic method used is not highly sensitive for the detection of Taeniid eggs. For this reason, other available methods are more frequently used, see Mathis A, Deplazes P, Eckert J. An improved test system for PCR-based specific detection of Echinococcus multilocularis eggs. Journal of Helminthology. 1996;70(3):219-222. doi:10.1017/ S0022149X00015443. This needs to be discussed.

Reply: We thank the Reviewer for this valuable comment. We modified the M&M – Parasitological analysis section, the Results, and the Discussion to provide a clearer and more detailed description of our testing procedures.

Before performing the parasitological and virological analyses, we attempted to genotype all samples to identify the host species and individual wolves. As described in the manuscript, this process had a success rate of approximately 50%, and we successfully identified 81 unique genotypes (that means 81 individuals) from 132 samples (the dog sample was excluded). The other 51 samples successfully genotyped were second or third samples belonging to the same individual and therefore we selected the most recently collected sample for each identified individual.

Subsequently, we performed a real-time duplex PCR on all successfully genotyped samples (n = 80), regardless of their copromicroscopic results, as well as on all samples positive for taeniid eggs at copromicroscopy (n = 24), in an attempt to roughly assess the prevalence (percentage of positive among individuals; i.e., among genotyped samples) and to increase the probability to detect the parasite if circulating (including not genotyped samples). This approach followed the method proposed by Obber et al. (2022; https://doi.org/10.1371/journal.pone.0268045), who demonstrated that qPCR directly applied to feces is a recommended strategy and allows a reduction in the number of samples required to detect infection. Based on previous research in the same area (Poglayen et al., 2017; http://dx.doi.org/10.1016/j.ijppaw.2017.01.001), we expected a low prevalence of Echinococcus spp. (about 5%), and a sample size of around 100 was deemed sufficient to detect the parasite at an expected prevalence of about 5% or more.

R#1 Line 155: Please, replace cysts with (oo)cysts

Reply: We replaced cycts with (oo)cysts.

R#1 Line 280: "Taeniidae is the most relevant parasitic genus"; Taeniidae is a parasitic family and not a parasitic genus

Reply: We corrected the term as suggested.

R#1 Lines 284-288: Having used a technique that is not highly sensitive for the detection of Taeniids, even the use of PCR can give false results if performed only on samples that tested positive to copromicroscopy

Reply: As noted in our response above, we revised the text accordingly to enhance the clarity and transparency of our methodological description.

R#1 Line 296: Please, replace taeniids with taeniid

Reply: We replaced taeniids with taeniid as requested.

R#1 Lines 280-359: In the discussion section, parasitological data should also be compared with those obtained in the following recent studies:

1. Cafiero SA, Petroni L, Natucci L, Tomassini O, Romig T, Wassermann M, Rossi C, Hauffe HC, Casulli A, Massolo A. New evidence from the northern Apennines, Italy, suggests a southward expansion of Echinococcus multilocularis range in Europe. Sci Rep. 2025 Mar 1;15(1):7353. doi: 10.1038/s41598-025-91596-7

2. Minichino A, Ciuca L, Dipineto L, Rinaldi L, Montagnaro S, Borrelli L, Fioretti A, De Luca Bossa LM, Garella G, Ferrara G. Exposure to selected pathogens in wild mammals from a rescue and rehabilitation center in southern Italy. One Health. 2025 Apr 21;20:101049. doi: 10.1016/j.onehlt.2025.101049

3. Cafiero SA, Petroni L, Natucci L, Casale L, Raffaelli M, Baldacci D, Di Rosso A, Rossi C, Casulli A, Massolo A, Hauffe HC, Perrucci S. Parasite diversity in grey wolves (Canis lupus) from Tuscany, central Italy: a copromicroscopical investigation. Int J Parasitol Parasites Wildl. 2025 May 31;27:101092. doi: 10.1016/j.ijppaw.2025.101092

Reply: We thank the Reviewer for these suggestions. We have added references (1) and (3) in the Discussion section. Reference (2) was not included, as we consider its epidemiological relevance to be limited, given that only four wolves were screened.

R#1 Lines 368-369: This sentence should be deleted or changed according to comments reported above

Reply: We thank the Reviewer for this comment. We have added a sentence highlighting that, although copromicroscopy can be considered a valuable tool for monitoring, its low sensitivity toward certain groups of parasites (such as taeniids) should be taken into account when interpreting the results.

R#1 Line 380: Please, replace Reference with References

Reply: We corrected accordingly.

R#1 Lines 381-553: In the references, parasite and animal genera and species should be written in italics

Reply: Text was edited accordingly.

Reviewer #2:

R#2 Line 43 - What type of legal protection?

Reply: We added in text “Habitat Directive 92/43/EEC and Italian Law 157/1992” to clarify the legal protection.

R#2 Lines 50-51 - Include an example of a zoonotic disease that corresponds to this assertion.

Reply: We thank the Reviewer for this suggestion. We have added an example of a pathogen that may threaten wolf conservation, canine distemper virus, and one that is relevant to public health, for which wolves may act as spreaders, Echinococcus granulosus.

R#2 Line 82 - need scientific name of red fox

Reply: The scientific name was included.

R#2 Line 88 - Echinococcus granulosus sense strict includes G1-G3 but just G1 is mentioned here, what about G2 and G3?

Reply: We corrected the genotypes of Echinococcus granulosus sensu stricto adding “(G1-G3)”.

R#2 Line 96 - wolf should be wolves

Reply: We corrected with wolves.

R#2 Line 158-159 - which unique genotypes are tested for?

Reply: We thank the Reviewer for this comment that helped us to clarify the genetic analysis performed. We modified the description in the Materials and Methods – Sample collection section, specifying that genetic analyses were performed to determine the host species and individual identity of wolves. In addition, we revised the Parasitological analysis section to further clarify this aspect. The detailed analysis of wolf genotypes is beyond the scope of the present study, as it is fully reported in another article by Dissegna et al. (2023), as indicated in the manuscript.

R#2 Line 287 - again which genotypes were test for?

Reply: Following the comment above and the comments of Reviewer 1, we also modified this phrase: “Following the method proposed by Obber et al. (2022), to increase the probability of detecting E. granulosus and E. multilocularis, we performed a method with high sensitivity (i.e., a real-time duplex PCR) on all samples attributable to individual wolf genotypes (n=80) - regardless of their copromicroscopic results - as well as on the samples belonging to unidentified wolves that tested positive for taeniid eggs at copromicroscopy (n=24).”

R#2 Generally an interesting and well written paper. Are Parvo and Coronavirus really the only two viruses of concern between feral/domestic dogs and wolves? Perhaps dive a little more into why these two were chosen and their history in the area, in contrast to other transmissible viruses of concern.

Reply: We thank the reviewer for this valuable suggestion. We selected canine parvovirus (CPV) and canine Ccoronavirus (CCoV) as target pathogens because both have been repeatedly reported in Italian wolf populations, including previous CPV cases documented in the Foreste Casentinesi National Park. These viruses are highly contagious, environmentally stable, and can persist in fecal material, making them ideal candidates for detection through copromicroscopy. Additionally, CPV and CCoV are well-known examples of pathogens that circulate at the interface between domestic dogs and wild carnivores, representing effective indicators of dog–wolf interactions. Other viral agents such as canine distemper virus (CDV), adenoviruses, and rabies virus are also relevant; however, their detection typically requires blood or tissue samples, which were not available in our non-invasive sampling design. We added a phrase in the M&M – virological analysis section to clarify these aspects.

---

## [Editor Report · Decision Letter 1]

1 Dec 2025

Monitoring the Health of Wolves (*Canis lupus*): Integrating Conservation and Public Health

PONE-D-25-46208R1

Dear Dr. Ferraro,

We’re pleased to inform you that your manuscript has been judged scientifically suitable for publication and will be formally accepted for publication once it meets all outstanding technical requirements.

Kind regards,

Ulrike Gertrud Munderloh, Ph.D.

Academic Editor

PLOS ONE
---

## [Editor Report · Acceptance letter]

PONE-D-25-46208R1

PLOS One

Dear Dr. Ferraro,

I'm pleased to inform you that your manuscript has been deemed suitable for publication in PLOS One. Congratulations! Your manuscript is now being handed over to our production team.

Kind regards,

on behalf of

Dr. Ulrike Gertrud Munderloh

Academic Editor

PLOS One